# The Current State of Zooplankton Diversity in the Middle Caspian Sea during Spring

Moldir Aubakirova [1,*], Zhanara Mazhibayeva [1], Saule Zh. Assylbekova [1], Kuanysh B. Isbekov [1], Bekzhan Barbol [2,3], Zamira Bolatbekova [1,4], Nurgul Jussupbekova [3], Aidana Moldrakhman [1] and Gulmira Satybaldiyeva [5]

1   Laboratory of Hydrobiology, Fisheries Research and Production Center, Almaty 050016, Kazakhstan
2   Faculty of Biology and Biotechnology, Al-Farabi Kazakh National University, Almaty 050040, Kazakhstan
3   Institute of Zoology, Almaty 050060, Kazakhstan
4   Faculty of Bioresources and Technology, Kazakh National Agrarian Research University, Almaty 050010, Kazakhstan
5   Department of Ecology, Saken Seifullin Kazakh AgroTechnical Research University, Astana 010011, Kazakhstan
*   Correspondence: judo_moldir@mail.ru

**Abstract:** The study of planktonic animals of the Caspian Sea is topical and, during the last centuries, has brought and continues to bring new results. This is an inevitable process attributed to the introduction of non-indigenous predatory representatives of zooplankton by ballast water of ships. During the study period, the zooplankton of the Middle Caspian Sea was represented by 13 taxa and consisted mainly of non-indigenous species typical of the Palearctic region. Native fauna was represented by three species—cladoceran *Evadne anonyx*, *Podonevadne camptonyx*, and copepod *Halicyclops sarsi* during the study period. The quantitative variables of zooplankton did not reach a high level in May 2020 and 2021 compared to the data of previous years. Cladocerans *Podonevadne camptonyx*, copepods *Acartia tonsa*, and larvae of Cirripedia dominated in 2020. By 2021, the dominant species of last year had been replaced by the cladoceran *Evadne nordmanni*. One-way ANOVA analysis detected significant differences in the quantitative variables of cladocerans and copepods between different years of zooplankton study. The decreasing abundance, biomass, and alteration of the dominant zooplankton species in the Middle Caspian Sea during the study period may be associated with the feeding type of predator species and a slight elevation of water salinity.

**Keywords:** zooplankton; non-indigenous species; predator; food base; copepoda; cladocera

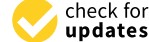



## 1. Introduction

Representatives of zooplankton occupy a key position in the aquatic food web and play a major role in energy transfer to macroinvertebrates or fish [1]. In particular, planktonic animals of the Caspian Sea are an important link in the ecosystem of the sea and are included in the diet requirements of benthic animals [1]. Sturgeons of the Caspian Sea are benthivorous [2], with about 90% of the world's sturgeon catch currently being carried out in the Caspian Sea [3]. Furthermore, the Caspian Sea is a hotspot of biological diversity [3–6]. Biodiversity conservation is an important direction of life science, and its significance has formed the basis of many international documents, including the Convention on Biological Diversity [7]. Therefore, the study of zooplankton assemblages in the Caspian Sea is topical and has brought and continues to bring new results regarding both species and taxocenosis levels during the last centuries [8]; an inevitable process attributed to the introduction of non-indigenous predatory species and increased anthropogenic pressure.

The studies of zooplankton in the Caspian Sea began at the end of the 19th century [9,10]. According to the literature data, zooplankton of the Caspian Sea was previously represented by 74 marine and brackish-water species, including 32 rotifers, 24 cladocerans,

and 18 copepods [11,12], and taking into account the freshwater forms that inhabit in the near-delta areas, the number of rotifer species reached up to 300 [13], cladocerans to 43, copepods to 50 species. Zooplankton of the Caspian Sea was represented by autochthonous species (16 species of cladocerans, 7 species of copepods, and 2 species of rotifers). The widespread endemic of the Caspian Sea was microcrustaceans *Eurytemora grimmi* (G. O. Sars, 1897), *Eurytemora minor* (Behning, 1938), *Polyphemus exiguus* G.O. Sars, 1897, and all species of *Apagis* and *Cercopagis*, besides *Cercopagis pengoi* (Ostroumov, 1891). They were widely distributed in the middle deep-water part of the Caspian Sea where there was stable salinity [14]. However, after functioning shipping links in the Caspian Sea were established with the Baltic and White Seas through the Volga–Baltic waterway and the White Sea–Baltic Channel, non-indigenous species started to transfer through ship ballast waters—one of the main reasons for the long-term change in marine ecosystems. A vivid example is the appearance of the predatory ctenophore *Mnemiopsis leidyi* (A. Agasis) in the Caspian Sea, which has influenced all links of the food web [11,12,15,16]. First of all, with the appearance of ctenophores, native species of the Caspian Sea, the microcrustaceans *Eurytemora grimmi* and *Eurytemora minor* began to disappear [11,12,15–17].

The last time scrutinizing zooplankton studies were carried out in the Caspian Sea was in 2008 and 2016. According to published data, 37 taxa were identified in the zooplankton of the Kazakhstan part of the Caspian Sea in that period [5]. The number of planktonic invertebrates varied from 3600 to 150,800 individual/m$^3$, and the biomass from 488.68 to 1766.5 mg/m$^3$. The basis of zooplankton quantitative variables was formed by rotifers *Brachionus quadridentatus*, cladocerans *Evadne anonyx*, *Podonevadne angusta*, *Podonevadne camptonyx*, *Podonevadne trigona*, and copepods *Acartia tonsa* and *Calanipeda aquedulcis*. In terms of abundance, the Shannon diversity index varied from 0.44 to 1.95 bits/ind. and from 0.51 bits/mg to 2.35 bits/mg [5,6].

Zooplankton of the Caspian Sea is relatively well studied; however that information outdated. The novelty of this current work is associated with assessing the current state of the zooplankton community of the Middle Caspian Sea and determining the possible reasons for zooplankton structural alterations.

## 2. Materials and Methods

### 2.1. Description of Study Area

The study of zooplankton in the middle part of the Caspian Sea was carried out in May 2020 and 2021 (Figure 1). Eight stations were installed to study the distribution of planktonic invertebrates in different depths of the sea.

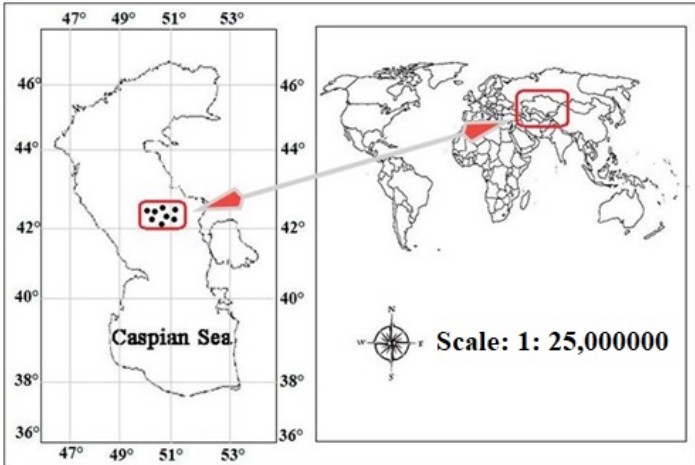

**Figure 1.** Map–scheme of the location of the sampling station (black dots in red square) in the middle part of the Caspian Sea, May 2020, 2021.

The station coordinates were determined using a GPS navigator (Garmin, Ltd., Olate, KS, USA) (Table 1). The temperature and salinity values were determined at each sampling site using Horiba U-50 equipment (Horiba, Ltd., Kyoto, Japan).

**Table 1.** The station coordinates and depths, May 2020, 2021.

| Station | Depth, m | Coordinates | |
|---|---|---|---|
| | | Longitude | Latitude |
| 1 | 97 22 | 51°48.3535′ E | 42°07.2959′ N |
| 2 | 97 22 | 51°48.3535′ E | 42°07.5659′ N |
| 3 | 97 22 | 51°48.3535′ E | 42°07.836′ N |
| 4 | 97 22 | 51°47.6347′ E | 42°08.106′ N |
| 5 | 97 22 | 51°47.9941′ E | 42°08.106′ N |
| 6 | 97 22 | 51°44.7594′ E | 42°10.8065′ N |
| 7 | 97 22 | 51°51.9476′ E | 42°10.8065′ N |
| 8 | 97 22 | 51°51.9476′ E | 42°05.4055′ N |

The Caspian Sea is the largest inland water body on the earth, with an area of 390,000 km$^2$. It lies in a moderately continental and warm continental climate zone with an average annual precipitation of up to 200–400 mm and a temperature gradient from −16 °C to +43 °C. Its main tributary is the transboundary River Volga. According to the morphological structure and physical and geographical conditions, the sea can be divided into three parts: the Northern, Middle, and Southern Caspian. The greatest depths of the northern, middle, and southern parts of the sea are 25, 788, and 1025 m, respectively. Due to a combination of physicochemical and biological characteristics of the waters in the Caspian Sea, the following water masses were distinguished: the North Caspian, the Upper Caspian, the deep Middle Caspian, and the deep South Caspian water masses. The North Caspian water mass occupies the northern part of the sea, and its volume is insignificant—less than 1% of the total volume of the sea. The temperature varies widely from 0 °C in winter to 25 °C in summer. In summer, most North Caspian water is warmed up from the surface to the bottom and has a temperature above 23–24 °C. In winter, most of the North Caspian is covered with ice; the water under the ice is almost equal to the freezing temperature. The Ural and Volga rivers flow into this part of the sea, with the salinity varying from 0.1–0.2 to 3–4‰, and the oxygen content from 2.0–3.5 mL/L. In forming the upper Caspian water mass, the primary role is played by the processes of winter cooling and mixing and summer heating, as well as dynamic processes in the upper layer of the sea (waves, wind currents, surge phenomena, internal waves). The depth of distribution of the winter vertical circulation determines the lower boundary of this water mass; it is located in the Middle Caspian in the 150–200 m layer, while in the Southern, it is 50–150 m. The upper Caspian water mass salinity is 12.7–13.0‰. This water mass has high oxygen content: in the upper layer—from 7.5–8.0 in winter to 6.0–6.5 mL/L in summer. Deep water masses are formed mainly in the winter due to the density runoff of cold waters from the northern regions of the sea and the eastern shelf. Deep Caspian water masses have the following average thermohaline characteristics: Middle Caspian (250–300 m, bottom)—temperature 3.9–5.2 °C, salinity 12.7–13.0‰, and oxygen content 3.0–5.5 mL/L; South Caspian (100–150 m, bottom)—temperature 5.7–6.3 °C, salinity 12.8–13.1‰, and oxygen content 2.0–3.5 mL/L. Winds blowing over the sea cause drift currents, and the density unevenness of seawater causes convective currents. Runoff currents are formed

over the mouth spaces of the rivers flowing into the Caspian Sea under the influence of river runoff. With prolonged and robust northwest winds, a surge of water forms in the southern part of the sea. As a result, even before the wind stops, a compensatory current to the north arises. Strong southerly winds contribute to a rise in the water level in the northern part of the sea, from where the water, even before the change in wind direction, rushes to the south, strengthening currents along the coast of the middle part of the Caspian Sea [18].

The water temperature in the surface layers of the Middle Caspian Sea varied from 18 °C to 21.6 °C, in the deep layers from 15.6 °C to 18.2 °C in 2020. Salinity values in the surface layers ranged from 11.74‰ to 14.23‰ in deep water layers from 13.62‰ to 15.12‰. In May 2021, the water temperature in the surface layers fluctuated from 18 °C to 20.5 °C in the deep layers from 15.36 °C to 18.0 °C. The salinity values varied in the surface layers from 11.3‰ to 14.2‰ in the deep layers from 11.3‰ to 13.2‰.

*2.2. Laboratory Processing*

A total of 16 zooplankton samples were collected, of which 8 were from the upper layer, 0–22 m, and 8 were from the deep layer, 0–97 m. Zooplankton was sampled using the Juday plankton net (net with an inlet diameter of 12 cm and mesh size of 65 μm), by stretching it from the bottom to the surface. Filtered water was poured into 250 mL plastic bottles and fixed with 40% formalin to a final concentration of 4% [19]. Further processing of the collected samples was carried out in the laboratory.

Representatives of zooplankton were identified at the species level using a key to species identification [11,13,20–22]. Planktonic animals were investigated microscopically using an MBS-10M binocular stereoscopic microscope (Biomed, St. Petersburg, Russia) and a Zeiss Primo Star microscope (Carl Zeiss, Jena, Germany) at 40× and 100× magnifications.

The quantitative processing of samples was carried out using standard methods [19]. The calculation of planktonic invertebrates was carried out in a certain part of the sample, depending on their abundance. After thorough mixing, three parts of the sample were taken using a 1 mL punch pipette. In this subsample, all encountered individuals and the different age stages of certain species (the most numerous) were counted in the Bogorov counting chamber. Next, the sample was concentrated to half of the initial volume. Three sub-samples were taken from it, in which the non numerous species and different age stages were counted. The whole process was repeated again when the sample was concentrated to 25 cm$^3$. The number of individuals of rare species was found when viewing the whole sample. In copepods, adult females, females with egg sacs, males, copepodites at 1–3 and 4–5 development stages, and nauplii were counted and measured separately. The same procedure was conducted for cladocerans (females with eggs or juveniles in the brood pouch, sterile females, males, and juveniles). In order to calculate the individual mass of planktonic invertebrates (wet mass, mg), the formula for the relationship between mass and body length was used [23]. Next, the abundance of individuals and the weight index of all species were summarized for the main groups of organisms and the community as a whole.

The abundance and biomass of individual species and total zooplankton were calculated per 1 m$^3$ of the water column using the formula [19]:

$$N = \frac{n \times \left(\frac{V_1}{V_2}\right)}{V_3} \tag{1}$$

where N—abundance (individual/m$^3$), n—abundance of individuals in a portion (specimen), $V_1$ is the volume of dilution (cm$^3$), $V_2$ is the volume of sub-sample (cm$^3$), and $V_3$ is the volume of filtered water (m$^3$).

The volume of filtered water was calculated by the formula:

$$V_3 = h \times \pi r^2 \tag{2}$$

where V$_3$ is the volume of filtered water, h—is the depth of the captured layer, $\pi$ is a mathematical constant ($\pi \approx 3.14$), and r is the inner radius of the Juday net hole.

The calculation of the species' average individual mass was carried out as the total biomass divided by the total abundance of zooplankton [19].

A finding of the dominant species was carried out according to Lyubarsky's scale [24] with a modification: absolute dominants included species that accounted for more than 60% of the quantitative variables of the community, dominants 31–60%, subdominants 10–30%.

### 2.3. Statistical Analysis and Comparisons with Previous Studies

The calculation of standard deviation was performed in Excel using the "STDEV" function. Zooplankton community datasets from depths of 22 m and 97 m were used to calculate the standard deviation. The similarity of zooplankton species composition in different years was determined by the Bray-Curtis index in the Primer 5 program [25]. We used statistical methods for ease of perception of information about the change of zooplankton quantitative variables in different years. One-way ANOVA analysis was used to determine whether there were any statistically significant differences between quantitative variables of zooplankton in different years of study. One-way ANOVA analysis was performed in R studio software [26]. The Shannon–Wiener Index and Pielou's Index were calculated both based on the abundance and the biomass of species in the sample [25,27,28] using Primer 6 program.

Field sampling (stations, seasons, and depths) and laboratory processing (species identification and quantification methods) of the studies in 2016 [6] were consistent with those of the current research. All methods of the 2008 studies, except for sample collection, were the same as the current sampling and methods [5]. Samples were collected only from the surface (upper layer, 38.4 m) in 2008 [5].

## 3. Results

### 3.1. Taxonomic Composition of Zooplankton Communities

Zooplankton of the Middle Caspian Sea was represented by 13 taxa in the spring of 2020 and 2021. The recorded taxa consisted of 1 Rotifera, 5 Cladocera, 3 Copepoda, and 4 taxa from the group of others during the study period (Table 2). The maximum number of taxa (12) was identified in May 2021. A year earlier, there were only 7 taxa registered; the recorded taxa belonged to different faunistic complexes. Native fauna of the Ponto-Caspian basin was represented by 3 species—cladoceran *Evadne anonyx* (G. Sars), *Podonevadne camptonyx* (G.O. Sars), and copepod *Halicyclops sarsi* (Akatova) during the study period. The rest of the holoplankton species are representatives of zooplankton of the Palearctic region.

Cladocerans *Podonevadne camptonyx*, *Podon intermedius* (Lilljeborg), copepod *Acartia tonsa*, temporary inhabitants of the water column Cirripedia, and Ostracoda often occurred in zooplankton of the surveyed area in 2020. Cladoceran crustaceans *Podon intermedius*, *Evadne nordmanni*, and copepods *Acartia tonsa* were widespread by 2021.

**Table 2.** Frequency of occurrence (%) of planktonic animals in the Middle Caspian Sea, May 2020, 2021.

| Taxon | 2020 | | 2021 | |
|---|---|---|---|---|
| | **Deep Layer** | **Upper Layer** | **Deep Layer** | **Upper Layer** |
| Rotifera | | | | |
| *Synchaeta littoralis* (Rousselet) | | | 25 | 13 |
| Cladocera | | | | |
| *Evadne anonyx* (G. Sars) | | | 100 | 100 |
| *Evadne nordmanni* Lovén | | | 100 | 100 |
| *Pleopis polyphemoides* (Leukart) | 38 | 25 | 100 | 88 |
| *Podon intermedius* (Lilljeborg) | 75 | 100 | 50 | 50 |
| *Podonevadne camptonyx* (G.O. Sars) | 100 | 100 | | |

**Table 2.** *Cont.*

| Taxon | 2020 | | 2021 | |
|---|---|---|---|---|
| | **Deep Layer** | **Upper Layer** | **Deep Layer** | **Upper Layer** |
| Copepoda | | | | |
| *Acartia tonsa* (Dana) | 100 | 100 | 100 | 100 |
| *Calanipeda aquaedulcis* (Kritschagin) | | | 38 | |
| *Halicyclops sarsi* (Akatova) | | | 38 | 25 |
| Others | | | | |
| Bivalvia gen. spp. | 38 | 75 | 100 | 100 |
| Spionidae gen. spp. | | | 50 | |
| Cirripedia gen. spp. | 100 | 88 | 100 | 100 |
| Ostracoda gen. spp. | 63 | 100 | 100 | 100 |
| Total: 13 | 7 | 7 | 12 | 10 |

*3.2. Quantitative Variables of Zooplankton Communities*

The highest abundance and biomass of planktonic invertebrates were recorded in May 2020 (Table 3). There was a slight decrease in the quantitative variables of zooplankton by 2021. The lower distribution boundary of genera *Evadne* and *Podonevadne* representatives is usually at most 50–60 m. Therefore, the low abundance and biomass of the community were characteristic of the deep part of the sea in both years of the study. A minor abundance increase in a deep layer of the Middle Caspian Sea occurred due to the Rotifera development in 2021.

**Table 3.** Quantitative variables of zooplankton communities in the Middle Caspian Sea, May 2020, 2021 (average values with standard deviation).

| Year | 2020 | | 2021 | |
|---|---|---|---|---|
| **Abundance, Individuals/m$^3$** | | | | |
| **Group** | **Deep Layer** | **Upper Layer** | **Deep Layer** | **Upper Layer** |
| Rotifera | 0 | 0 | 5.50 ± 5.36 | 0.13 ± 0.13 |
| Cladocera | 813.75 ± 281.26 | 2092.11 ± 443.88 | 736.37 ± 166.97 | 948.50 ± 190.93 |
| Copepoda | 1021.88 ± 348.56 | 4071.51 ± 1032.34 | 679.13 ± 106.10 | 2816.50 ± 438.64 |
| Others | 156.38 ± 49.87 | 589.19 ± 111.29 | 726.25 ± 174.33 | 2815.75 ± 547.72 |
| Total | 1992.0 ± 574.40 | 6752.75 ± 1329.02 | 2147.25 ± 317.28 | 6580.88 ± 955.48 |
| **Biomass, mg/m$^3$** | | | | |
| Rotifera | 0 | 0 | 0.02 ± 0.02 | 0.0002 ± 0.0002 |
| Cladocera | 132 ± 43.40 | 310.69 ± 62.02 | 151.37 ± 37.26 | 175.56 ± 45.40 |
| Copepoda | 25.42 ± 9.31 | 95.28 ± 25.16 | 5.07 ± 1.08 | 22.20 ± 4.03 |
| Others | 1.72 ± 0.81 | 6.86 ± 2.23 | 4.07 ± 0.70 | 16.56 ± 2.10 |
| Total | 159.71 ± 50.75 | 412.82 ± 73.64 | 160.54 ± 37.22 | 214.32 ± 49.33 |

*3.3. Dominant Species in Zooplankton Communities*

Copepods dominated in abundance, and cladoceran prevailed on biomass in the zooplankton community of the Middle Caspian Sea (Table 4). The complex of dominants was constant in the sea's deep and surface parts. In 2020, the list of dominants consisted of three taxa—cladoceran *Podonevadne camptonyx*, copepod *Acartia tonsa*, and Cirripedia from the group of others. By 2021, the dominant species of previous years has been replaced by Cladocera *Evadne nordmanni*.

It is important to note that the population of *Acartia tonsa* was represented by individuals at the juvenile stages, copepodites at 1–3 and 4–5 development stages, and individuals at the nauplius stages. Copepodites of *Acartia tonsa* at 4–5 development stages contributed to the abundance of the population by approximately 14%; the proportion of copepodites at 1–3 development stages varied from 50 to 53%, as for individuals at nauplius stages, this value was between 14–15%. The latter explained the small proportion of copepod *Acartia tonsa* in biomass.

**Table 4.** Composition of dominant species in zooplankton of the Middle Caspian Sea, May 2020, 2021.

| Group | Species | Deep Layer | Upper Layer | Deep Layer | Upper Layer |
|---|---|---|---|---|---|
| | | 2020 | | 2021 | |
| | | Abundance, % | | | |
| Cladocera | *Podonevadne camptonyx* | 39.82 | 29.46 | 0 | 0 |
| | *Evadne nordmanni* | 0 | 0 | 34.0 | 13.20 |
| Copepoda | *Acartia tonsa* | 51.30 | 60.29 | 31.59 | 42.78 |
| Others | *Cirripedia* | 5.30 | 4.72 | 27.48 | 35.32 |
| | | Biomass, % | | | |
| Cladocera | *Podonevadne camptonyx* | 82.22 | 74.45 | 0 | 0 |
| | *Evadne nordmanni* | 0 | 0 | 94.05 | 78.03 |
| Copepoda | *Acartia tonsa* | 15.92 | 23.08 | 3.16 | 10.35 |

### 3.4. Species Number and Diversity Indices of Zooplankton Communities

The Shannon–Wiener diversity index values ranged from 1.29 to 1.78 bits/ind and from 0.56 to 1.77 bits/mg (Table 5) during the study period. The highest Shannon diversity index was recorded in 2021. The average values of the Pielou evenness index in 2020 and 2021 were almost similar.

**Table 5.** Diversity indices of zooplankton communities of the Middle Caspian Sea, May 2020, 2021.

| Variables | 2020 | | 2021 | |
|---|---|---|---|---|
| | Deep Layer | Upper Layer | Deep Layer | Upper Layer |
| Shannon–Wiener Index (H) by abundance | $1.29 \pm 0.10$ | $1.41 \pm 0.09$ | $1.78 \pm 0.08$ | $1.74 \pm 0.10$ |
| Shannon–Wiener Index (H) by biomass | $0.67 \pm 0.13$ | $0.95 \pm 0.08$ | $0.56 \pm 0.10$ | $1.77 \pm 0.15$ |
| Pielou's Index (J) | $0.56 \pm 0.04$ | $0.55 \pm 0.03$ | $0.43 \pm 0.05$ | $0.59 \pm 0.03$ |

### 3.5. Statistical Analysis and Comparisons with Previous Studies

#### 3.5.1. Taxonomic Composition of Zooplankton Communities

The last time zooplankton studies in the Caspian Sea were carried out was in 2008 and 2016 [5,6]. The number of zooplankton taxa in the Middle Caspian Sea diminished from 21 in 2008 to 9–12 in 2016 and 2020–2021 [5,6]. According to the cluster analysis based on the values of the Bray–Curtis index (Figure 2), the resemblance of zooplankton species composition of more than 50% was found between all studied years and depths. The registered cladocerans from 2016, *Podon polyphemoides* Leukert and *Podonevadne camptonyx podonoides* G.O. Sars, fell out of zooplankton of the Middle Caspian Sea in 2020 and 2021. Furthermore, the species composition of zooplankton in 2020 and 2021, compared with those of 2008, showed through comparative analysis that Rotifers *Brachionus quadridentatus* Hermann, 1783; *Synchaeta cecilia* Rousselet, 1902; *Synchaeta stylata* Wierzejski, 1893; cladocerans *Cornigerius maeoticus hircus* (G.O. Sars, 1902), copepods *Idyaea furcata* (Baird, 1837), Ergasilidae gen.spp., Calanoida gen.spp., and temporary inhabitants of the water column larvae of *Hediste diversicolor* (O.F. Müller, 1776), Nematoda gen.spp., found in 2008, had fallen out of the zooplankton in 2020 and 2021 [5,6].

#### 3.5.2. Quantitative Variables of Zooplankton Communities

One-way ANOVA analysis revealed significant differences in cladoceran and copepod abundance in different years of study ($p > 0.05$) (Table 6). The abundance of cladocerans in 2020 ($813.75 \pm 281.26$ individuals/m$^3$ deep layer; $2092.11 \pm 443.88$ individuals/m$^3$ upper layer) and 2021 ($736.37 \pm 166.97$ individuals/m$^3$ deep layer; $948.50 \pm 190.93$ individuals/m$^3$ upper layer) decreased compared to 2016 ($3052 \pm 1758$ individuals/m$^3$) [5,6]. The abundance of copepods in 2021 ($679.13 \pm 106.10$ individuals/m$^3$ deep layer; $2816.50 \pm 438.64$ individuals/m$^3$ upper layer) was at a low level compared to previous years [5,6].

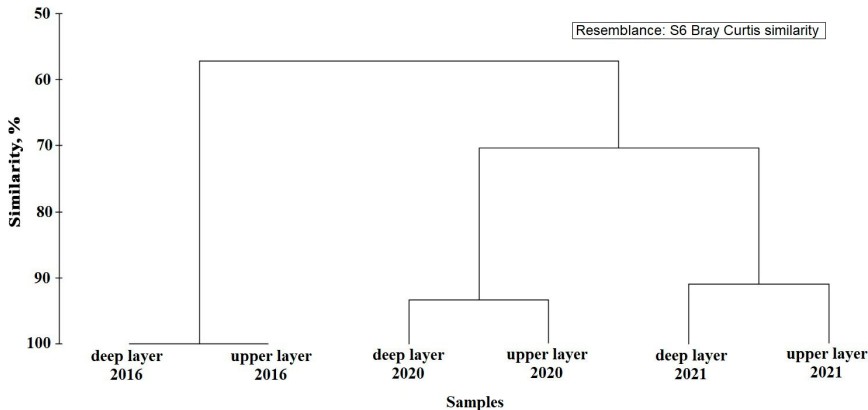

**Figure 2.** A dendrogram of the similarity of zooplankton species composition in different periods of study of Middle Caspian Sea.

**Table 6.** Results of one-way ANOVA by the abundance of zooplankton communities of the Middle Caspian Sea in different years.

| Variable | Df | Sum of Squares | Square Mean | F Value | Significance |
|---|---|---|---|---|---|
| Rotifera | 1 | 7.000 | 7.000 | 0.993 | 0.424 |
| Cladocera | 1 | 19,256,014 | 19,256,014 | 97.629 | 0.0101 * |
| Copepoda | 1 | 533,053,889 | 533,053,889 | 78.396 | 0.01025 * |
| Others | 1 | 812,944 | 812,944 | 0.574 | 0.528 |
| Total | 1 | 805,222,276 | 805,222,276 | 53.261 | 0.0183 * |

Notes. Significance codes: 0.01 '*'.

One-way ANOVA analysis found significant differences in copepod biomass and in total zooplankton biomass in different years of study ($p > 0.05$) (Table 7). The biomass of copepod ($5.07 \pm 1.08$ mg/m$^3$ upper layer; $22.20 \pm 4.03$ mg/m$^3$ deep layer) in 2021 decreased significantly compared to the data of 2016 ($316 \pm 214$ mg/m$^3$) and slightly to that of 2008 and 2020 [5,6]. The biomass of zooplankton in 2020 and 2021 increased compared to 2008 ($20.2 \pm 10.7$ mg/m$^3$) [5] but was low in contrast to 2016 ($392 \pm 257$ mg/m$^3$) [6]. A sharp increase in biomass occurred due to the rising of cladoceran biomass in 2021.

**Table 7.** Results of one-way ANOVA by biomass of zooplankton communities of the Middle Caspian Sea in different years.

| Variable | Df | Sum of Squares | Square Mean | F Value | Significance |
|---|---|---|---|---|---|
| Rotifera | 1 | <0.0001 | <0.0001 | <0.0001 | <0.0001 |
| Cladocera | 1 | 9148 | 9148 | 1.373 | 0.362 |
| Copepoda | 1 | 206,531 | 206,531 | 202.212 | 0.00491 ** |
| Others | 1 | 108.67 | 108.67 | 2.568 | 0.250 |
| Total | 1 | 151,373 | 151,373 | 29.408 | 0.0324 * |

Notes. Significance codes: 0.001 '**' 0.01 '*'.

### 3.5.3. Dominant Species in Zooplankton Communities

In 2020 and 2021, the composition of the dominant planktonic invertebrates in the surveyed area changed compared to the data of previous years [5,6]. Rotifera and copepods dominated in terms of abundance, and copepods prevailed regarding biomass in 2008 [5]. Copepods developed the quantitative variables of zooplankton in 2016, which accounted for 56% (abundance) and 97% (biomass) [6]. The rest of the proportion in quantitative variables belonged to cladocerans (mainly to *Evadne anonyx*) in 2016.

The constant dominant of previous years was copepod *Acartia tonsa*, the proportion of which reduced from 60–97% in 2008, 2016, and 2020 to 40% in 2021 [5,6]. The updated complex of dominants included the cladoceran *Evadne nordmanni* in 2021.

### 3.6. Diversity Indices of Zooplankton Communities

One-way ANOVA analysis detected significant differences only in the Shannon–Wiener diversity Index by abundance in different years of study ($p > 0.05$) (Table 8). The Shannon–Wiener Diversity Index calculated based on the abundance and biomass of zooplankton increased significantly from 0.58 to 1.78 bit/ind., and from 0.26 to 1.77 bit/mg compared to 2016 data. The comparative analysis did not reveal significant increases in zooplankton diversity indices with 2008 data.

**Table 8.** Results of one-way ANOVA by diversity indices of zooplankton communities of the Middle Caspian Sea in different years.

| Variable | Df | Sum of Squares | Square Mean | F Value | Significance |
|---|---|---|---|---|---|
| Shannon–Wiener Index (H) by abundance | 1 | 1.8309 | 1.8309 | 96.207 | 0.0102 * |
| Shannon–Wiener Index (H) by biomass | 1 | 0.5212 | 0.5212 | 4.824 | 0.159 |

Notes. Significance codes: 0.01 '*'.

## 4. Discussion

During the study period, the Middle Caspian Sea zooplankton had low species richness, abundance, and biomass values; this zooplankton structure is characteristic of deep water [29]. The content of nutrients in deep water areas is lower than in estuaries; therefore, conditions for the growth and development of aquatic organisms are unfavorable in deep water areas [30].

One-way ANOVA analysis, which allows the determination of statistically significant differences between structures of zooplankton in different years of study, detected significant differences in the abundance of cladocerans, copepods, copepod biomass, and Shannon–Wiener diversity Index. The abundance and biomass of crustaceans (cladocerans and copepods) decreased in 2021 compared to 2008, 2016, and 2020 data. Alterations in the structure of the community may be associated with fluctuations in environmental parameters (primarily salinity, temperature) and competition for food and with the presence and impact of predators [15,16,20].

As in the Black and Azov Seas, in the Caspian Sea, the maximum value of the quantitative variables of crustaceans, which play the dominant role, is typical for the summer-autumn period [29,31]. Nevertheless, copepods *Acartia tonsa* play a dominant role in zooplankton of the Caspian Sea throughout the whole year [5,6]. The contribution of this species in abundance and biomass of zooplankton of the Caspian Sea reaches 60–90%. The proportion of *Acartia tonsa* in the quantitative variables of zooplankton of the Middle Caspian Sea reduced from 60–97% in 2008, 2016, and 2020 to 40% in 2021 [5,6]. In 2021, the dominant species of zooplankton was cladoceran *Evadne nordmanni*. According to the literature, *Acartia tonsa* is a euryhaline and eurythermal species. It can withstand salinity fluctuations from 10 to 15‰ and temperatures from 5 °C to 33 °C [32]. According to hydrochemical data, salinity and water temperature were optimal for copepod *Acartia tonsa* in 2021. Another reason for decreasing the quantitative variables of the constant dominant copepod *Acartia tonsa* should be competition over common food resources. The life cycle of crustaceans *Acartia tonsa* strictly depends on the amount of the available feeding base. The low amount of forage supplies cease the growth rate of these crustaceans [33]. The forage base of cladoceran *Evadne nordmanni* and copepod *Acartia tonsa* have similar elements. Diatoms, dinoflagellates, and peredinum are part of the food base of these species [20].

Due to a lack of forage copepods *Acartia tonsa* may be prey for predatory cladoceran *Evadne nordmanni*. This is supported by the presence of copepod eggs in the food base of *Evadne nordmanni* in water bodies of Scotland [34], as well as the case that *Evadne nordmanni* only consumes food of animal origin in the Mediterranean Sea [35]. Additionally, the forage of other representatives of the genus *Evadne* or congeneric residents consisted of 50% *Eurytemora*, 22% of copepod nauplii, and 28% of polyphemids in the Caspian Sea [11].

During the research period, *Acartia tonsa* was represented mainly by individuals at the initial stages of development. It is known that in the planktonic fauna of the Middle Caspian Sea in other periods of 2021, along with the predatory cladoceran *Evadne nordmanni*, there was a predator *Mnemiopsis leidyi* [36]. It is possible that ctenophore *Mnemiopsis leidyi* started to feed on adult individuals of copepod *Acartia tonsa*. However, contrary to other prey copepods of ctenophore, *Acartia tonsa* can save its population from ctenophore *Mnemiopsis leidyi* due to its biological characteristics [37]. Copepod *Acartia tonsa* does not carry eggs attached to the genital segment sac and spawns eggs directly into the water. In this case, some eggs fall to the bottom, forming a resting stage. Therefore, the cessation of the lifecycle of an egg-carrying female does not lead to the cessation of all *Acartia tonsa* offspring. The complete disappearance of the adult individuals of *Acartia tonsa*, associated with the intensive development of the ctenophore, does not influence copepodite at an early stage, and nauplii constantly occur in the plankton.

Except for *Acartia tonsa*, cladocerans *Evadne anonyx* formed 10% of the zooplankton biomass in 2008 [5] and 20% of zooplankton quantitative variables in 2016 [6]. However, it was completely absent in the spring of 2020. *Evadne anonyx* was present in zooplankton in the spring of 2021 but did not attain a high population abundance. Its abundance peaks occur from July to October at 16–20 °C temperatures and salinity of 12–13‰ in the Caspian Sea [5]. The salinity and temperature are essential factors controlling the seasonal population dynamics of cladoceran *E. anonyx*. A slight change in salinity could be the reason for the decrease in the contribution of *E. anonyx* to the quantitative variables of zooplankton. The water salinity was 12–14‰ in 2016, favoring the cladocerans [6,38].

The Shannon's diversity index calculated based on zooplankton abundance and biomass increased significantly from 0.58 to 1.78 bit/ind., and from 0.26 to 1.77 bit/mg compared to 2016 data. The similar values of Shannon's diversity index were noted during the short-term elevation of salinity and organic pollution in the Caspian Sea [5] and in the Azov Sea [29].

## 5. Conclusions

Thus, 13 zooplankton taxa were identified in the Middle Caspian Sea. Zooplankton of the Middle Caspian Sea consisted mainly of non-indigenous species typical of the Palearctic region. Native fauna was represented by three species—cladoceran *Evadne anonyx*, *Podonevadne camptonyx*, and copepod *Halicyclops sarsi* during the study period.

One-way ANOVA analysis detected significant differences in the quantitative variables of cladocerans and copepods between different years of zooplankton study. The decreasing abundance, biomass, and alteration of dominant zooplankton species in the Middle Caspian Sea during the study period may be associated with the feeding type of predator species and a slight elevation of water salinity.

**Author Contributions:** Conceptualization, M.A.; methodology, M.A.; software, M.A.; validation, M.A.; formal analysis, M.A.; investigation, M.A.; resources, M.A.; data curation, M.A.; writing—original draft preparation, M.A.; writing—review and editing, M.A.; visualization, S.Z.A., K.B.I., Z.M., B.B., Z.B., N.J., A.M. and G.S.; supervision, M.A.; project administration, K.B.I.; funding acquisition, K.B.I. All authors have read and agreed to the published version of the manuscript.

**Funding:** This research has is funded by the Science Committee of the Ministry of Science and Higher Education of the Republic of Kazakhstan (Grant No. AP09058158).

**Institutional Review Board Statement:** Not applicable.

**Informed Consent Statement:** Not applicable.

**Data Availability Statement:** No new data were created or analyzed in this study. Data sharing is not applicable to this article.

**Acknowledgments:** The authors sincerely thank G. Hörmann (Germany, University of Kiel) for teaching the basics of the R Software and other comprehensive bits of help.

**Conflicts of Interest:** The authors declare no conflict of interest.

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
