# Peer review of "The Current State of Zooplankton Diversity in the Middle Caspian Sea during Spring"

_diversity, doi:10.3390/d15070798_

Round 1

Reviewer 1 Report

Dear authors

The article is devoted to the description of the species composition of the zooplankton community in the Middle Caspian Sea. The text is written logically, consistently and interestingly. It is not clear why the authors included pictures and their description in the discussion. In the opinion of the reviewer, this part should definitely be transferred to the results, if necessary at all (see comments in the text).

Much of what is described in the text of this work has already been published in one form or another by the authors themselves or their colleagues. However, I hope there is some novelty in this study. As I understand it, the novelty lies in the description of the entire zooplankton community and its comparison with previous studies in 2008 and 2016, although these data are also published in the Waters journal of the same edition by Aubakirova et al. 2022. In any case, this is not enough for an article in a highly rated publication, since the work is based on 32 samples (without repetitions) from 8 stations in the Middle Caspian. Moreover, the list of species is scanty and includes only 13 taxa. I would recommend that you indicate in the text of the article what the authors see as the novelty of this study. It would be good to further strengthen the work (maybe take a couple more years for comparison or other sea regions, etc.) so that it differs more from the previous one.

Reasoning about the change in the abundance and biomass of groups in communities compared to previous years also raises questions. According to the pictures 3 and 4, no drastic changes are observed, except for an unexpected surge in the number of Cladocera in 2016. In fact, if we refer to the article by Kurochkina et al. 2022 Acarta tonsa dominated in abundance and biomass. An important conclusion of the work follows from this - since 2016, there has been a change not only in mass species in the studied point of the Caspian Sea, but even a change in the community as a whole - from copepod, it became cladoceran.

I hope the authors will be able to bring their work to an acceptable form in order to be published in this collection.

Best wishes

Author Response

Dear Reviewer,

We express our deep gratitude for your comments and suggestions regarding our manuscript. The manuscript has been substantially revised based on your suggestions and comments.

Kind Regards,

Moldir Aubakirova, Zhanara Mazhibayeva, Saule Zh. Assylbekova, Kuanysh B. Isbekov, Bekzhan Barbol, Zamira Bolatbekova, Nurgul Dzhusupbekova, Aydana Moldrakhman, Gulmira Satybaldieva

Reviewer 2 Report

diversity-2309253

The current state of zooplankton diversity in the Middle Caspian Sea

Moldir Aubakirova, et al.

The manuscript presents a study of the zooplankton community in the mddle part of the Caspian Sea, based on two surveys during May 2020 and May 2021. The major flaw is the serious sporadic linguistic inconsistencies, as well as, some reasoning inconsistencies in the Discussion, which should be rewritten or removed. In general, the Discussion cohesiveness needs improvement. Check carefully the English in the entire manuscript.

TITLE

At the end the title, add "during spring".

ABSTRACT

L20 "consists" use past tense. In general, during description use the same tense.

L22 Replace "copepoda" with "copepod".

INTRODUCTION

L39 Replace "forage base" with "diet requirements" and use a citation.

L41, 42 " Furthermore, the Caspian ....diversity" a citation is needed.

L42 Delete "As well-known".

L53 Replace "inhibit in" with "inhabit".

L56 Delete "i.e. species of Caspian origin" is repetition.

MATERIALS AND METHODS

L65, 65 Move "A total of 16 samples... the deep layer, 97 m" to the next section (zooplankton). Also insert information about the tows, do you mean 0-22 and 0-97?

L123, 124 More details on the identification of the copepod young stages (nauplii, early copepodites) are needed. Such as, do the species results correspond to adults only (Table 1.)? It is not clear if the numbers of nauplii and copepodites are included in the totals.

L133 A citation is missing.

RESULTS

L170, 171 "The maximum .... May 2021 – 12” unclear!

L176 - 178 Unclear!

L179, 180 Throughput text: use “copepod” (not “copepod” or “copepod crustaceans”) cladoceran (not “cladoceran crustaceans”).

L199, 200 This is unusual. The only explanation could be the extremely high numbers of very early stages. If this is the case, the authors should give a special emphasis on this point, modifying the below (in the paragraph) relevant comment (what about nauplii?).

DISCUSSION

L215, 216 When different years are compared, specify that you compare the corresponding spring period.

L 233 Replace "gen.sp." with "spp.".

L239- 245 I cannot understand your claim concerning the two works of 1968 and 1971 respectively. This is totally unreasonable to me. If you mean something else modify this part, otherwise remove it.

Discussion in general: there are repetitions of the explanations concerning the change in zooplankton community. When you use an explanation, already been referred, just use it in brief.

CONCLUSIONS

L322 - 324 "The quantitative variables of zooplankton .... of previous years."  to be removed, unnecessary

Author Response

(The authors gave the same response as above.)

Reviewer 3 Report

Review for the paper "The current state of zooplankton diversity in the Middle Caspian Sea during spring" by Moldir Aubakirova, Zhanara Mazhibayeva, Saule Zh. Assylbekova, Kuanysh B. Isbekov, Bekzhan Barbol, Zamira Bolatbekova, Nurgul Dzhusupbekova, Aydana Moldrakhman, Gulmira Satybaldieva submitted to "Diversity".

General comment.

Zooplankton assemblages constitute a significant component within aquatic ecosystems, functioning as a crucial intermediate link between primary producers and higher trophic levels. The primary focus of the paper is to delineate the composition, abundance, and biomass of zooplankton inhabiting the Caspian Sea. Despite a paucity of reports concerning the zooplankton present in the region, the current study could potentially pique the interest of scientists specializing in this field. Nonetheless, it is evident that the paper, in its current form, is not ideally suited for publication. The research conducted is predominantly descriptive in nature and thus does not test a specific hypothesis. Additionally, it lacks an analysis that probes the possible influence of environmental variables on the zooplankton community. Furthermore, the methodology employed for the collection of samples exhibits certain limitations, and the statistical treatment applied is insufficient for deriving comprehensive conclusions. Therefore, it is imperative that the authors rigorously revise their manuscript in order to improve the quality of the scientific content and enhance the presentation of the data.

Major points.

1. Introduction. A brief overview of recent zooplankton studies in the Caspian Sea must be included in the ms.

2. Introduction. The novelty of the study must be emphasized at the end of the Introduction.

3. Materials and Methods. Section 2.1. A table showing depths, coordinates of the stations and sampling dates should be provided as supplementary information.

4. Materials and Methods. Section 2.1. Environmental conditions of the region (climate, general hydrology, water masses, currents, and bathymetry) must be clearly described.

5. Materials and Methods. Section 2.2. There exists a significant concern pertaining to the sampling methodology of zooplankton, specifically in regards to the formula (L136) utilized to calculate the volume of water being filtered through the net. The net mesh of the Juday net employed in this research study has a dimension of 30 µm, which is an exceedingly fine net that may be easily clogged even in the absence of a phytoplankton bloom. Given that the tows involved in the study seemingly did not incorporate flowmeters, it can be deduced that the volume of water filtered would be markedly reduced, thereby posing a potential risk for zooplankton undersampling. Consequently, the calculations of total abundance and biomass may be skewed, leading to underestimations. It is imperative that this issue be addressed in the current section or deliberated upon within a special section to ensure the validity of the study findings.

6. Materials and Methods: Section 2.3 on Statistical treatment requires elaboration. Firstly, comparisons between zooplankton of different years and depth layers must be made using relevant criteria such as one-way ANOVA, Kruskal-Wallis test or others. Secondly, diversity indices such as Shannon-Wiener index and Pielou evenness should be calculated and compared for each sampling period. Lastly, I recommend performing cluster analysis to depict differences between sampling stations instead of a network analysis.

7. Discussion: Other possible factors that could explicate differences in the zooplankton structure and abundance must be considered, such as environmental and climatic influences, as well as anthropogenic changes. Specifically, significant environmental changes have been recorded in the Caspian Sea during the last decade, associated with the sea level, which could influence the plankton fauna.

8. Discussion: The authors must also compare their data with similar ecosystems to provide a comprehensive understanding of the subject.

9. Discussion: The ecological significance of the alterations in zooplankton composition on the ecosystem of the region must be discussed in detail.

10. References: The Latin names of the genera and species must be in italics. I highly recommend checking thoroughly the literature.

11. Finally, the English throughout the text should be revised.

Specific remarks.

L23 and below in the text. Cirripedia must be in ordinary font not in Italics.

L101. Indicate diameter of the net.

L108. Indicate magnification to observe zooplankton taxa.

L121. Provide references for relevant guides to identify zooplankton taxa.

L123. Clarify, in which units the biomass was expressed (wet, dry, carbon).

L149. Specify the measure of similarity (Euclidean distance, Bray-Curtis index or other).

L180. It is unclear what do mean the values in the Table (occurrence, frequency or other). Please, explain.

Table 1. Bivalvia, Spionidae, Cirripedia, Ostracoda must be in ordinary font not in Italics.

Table 3. Cirripedia must be in ordinary font not in Italics.

Section 3.3.2. The abundances should be rounded for better presentation (e.g. L238: 814±281 instead of 813.75±281.26).

Author Response

Dear Reviewer,

Thank you !

Round 2

Reviewer 1 Report

Dear authors

Thank you for the performed work on the paper. I find some improving compared to previous version. But I did not find the answer to more than half of my Comments and Suggestions for Authors. I still did not find something new in the paper comparing to previous published studies.  As a result, I find the paper is still not merit to be published in Diversity journal.

Author Response

Dear Reviewer,

We express our deep gratitude for your comments and suggestions regarding our manuscript. Revisions are highlighted with green colour.

Kind Regards,

Moldir Aubakirova, Zhanara Mazhibayeva, Saule Zh. Assylbekova, Kuanysh B. Isbekov, Bekzhan Barbol, Zamira Bolatbekova, Nurgul Dzhusupbekova, Aydana Moldrakhman, Gulmira Satybaldieva

Reviewer 3 Report

The authors have revised the paper according to my comments. 

Round 3

Reviewer 1 Report

Dear authors

If the editor suppose your paper merits to be published. Let’s publish